# Numerical Modelling of Structures Adjacent to Retaining Walls Subjected to Earthquake Loading

**Xiaoyu Guan and Gopal S. P. Madabhushi \***

Schofield Centre, Department of Engineering, University of Cambridge, Trumpington Street, Cambridge CB2 1PZ, UK; xg257@cam.ac.uk
**\*** Correspondence: mspg1@cam.ac.uk; Tel.: +44-12-2376-8053

**Abstract:** In an urban environment, it is often necessary to locate structures close to existing retaining walls due to congestion in space. When such structures are in seismically active zones, the dynamic loading attracted by the retaining wall can increase. In a novel approach taken in this paper, finite element-based numerical analyses are presented for the case of a flexible, cantilever sheet pile wall with and without a structure on the backfill side. This enables a direct comparison of the influence exerted by the structure on the dynamic behaviour of the retaining wall. In this paper, the initial static bending moments and horizontal stresses prior to application of any earthquake loading are compared to Coulomb's theory. The dynamic behaviour of the retaining wall is compared in terms of wall-top accelerations and bending moments for different earthquake loadings. The dynamic structural rotation induced by the differential settlements of the foundations is presented. The accelerations generated in the soil body are considered in three zones, i.e., the free field, the active and the passive zones. The differences caused by the presence of the structure are highlighted. Finally, the distribution of horizontal soil pressures generated by the earthquake loading behind the wall, and in front of the wall is compared to the traditional Mononobe-Okabe type analytical solutions.

**Keywords:** earthquake; finite element modelling; retaining walls; structures

## 1. Introduction

The increasing population in large cities and towns around the world place a high demand on available land. In such situations, excavations are required next to existing buildings to construct new metros, railways, highways or new buildings with basement floors. Retaining walls are often used to create the necessary difference in ground levels with backfills placed behind the retaining walls or excavations carried out in front of them. In the former case, new buildings could be constructed close to the retaining wall, while, in the latter case, where existing buildings are present behind the wall, the retaining walls have to protect from any damage due to ground movement. The location of these structures can often be quite close to the retaining wall, often only a few meters away. An example of a retaining wall close to an existing structure is shown in Figure 1. In this figure the retaining wall is anchored using soil nails, but many cases exist where cantilever retaining walls are used. The designer has to evaluate the additional soil-structure interaction effects due to the presence of the structure on the backfill side.

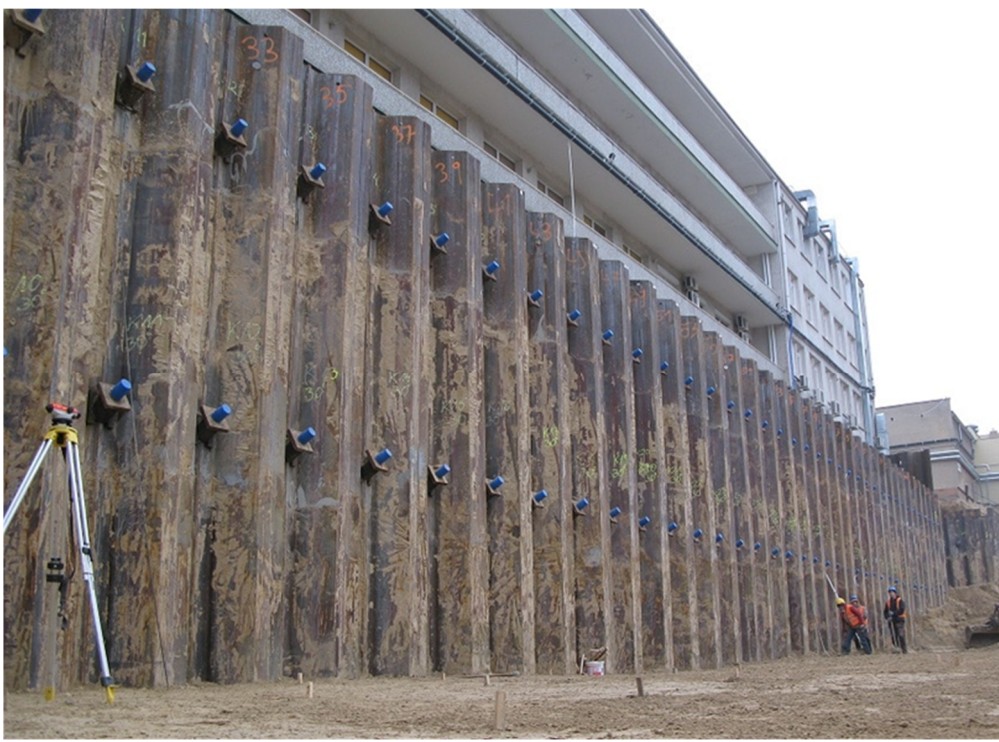

**Figure 1.** A structure in close proximity to a retaining wall [1] (courtesy Aarsleff Ground Engineering Ltd. (Newark, UK)).

In seismic regions of the world, additional loading is expected on the retaining wall due to earthquake loading. In these extreme loading cases, it can be anticipated that additional lateral loads act on the retaining wall due to increased horizontal stresses, additional inertial loads from the building, etc. There is a significant dynamic soil-structure interaction due to the action of earthquake loads. It is imperative that designers provide adequate wall sections for the retaining wall to withstand these additional loads. Additionally, the wall movements can have adverse effects on the foundations of the structure causing it to settle and rotate. The interaction between the retaining walls and structures is also important from the bridge engineering perspective. For example, the spandrel walls of masonry bridges, acting as retaining walls, interact strongly with the backfill soils. Pelà et al. [2] and D'Amato et al. [3] have investigated the seismic behaviour of masonry arch bridges. Similarly, the use of static and dynamic tests on bridges was recommended based on the modal analyses by Marcheggiani et al. [4]. In all these cases, the consideration of the interaction between the abutment walls and the soil backfill is important, particularly during strong earthquake loading.

There have been examples of damage to retaining walls in previous earthquakes—for example, a sea front retaining wall collapsed during the Bhuj earthquake in 2001 [5]. Zeng and Steedman [6] used dynamic centrifuge modelling to investigate flexible, cantilever retaining walls subjected to earthquake loading with dry and saturated backfills. Madabhushi and Zeng [7,8] investigated the same retaining walls using the finite element method in the time domain using a fully coupled code called Swandyne [9]. They show that the finite element (FE) code SWANDYNE was able to capture the dynamic response observed in the centrifuge tests both in terms of the wall displacements and dynamic bending moments. Cilingir et al. [10] extended this work to investigate anchored retaining walls in dry soils. Their research shows that SWANDYNE was able to capture the magnitude of the anchor forces recorded in the dynamic centrifuge test well, in addition to the bending moments and wall displacements. More recent improvements in the numerical modelling of retaining walls were carried out by Callisto and Soccodato [11] and Conti and Viggiani [12] with the aim of developing improved design methods. Conti and Viggiani [12] used the "threshold or yield acceleration" concept to estimate the lateral wall displacements and the accumulation of the wall displacements over different

cycles of an earthquake. Yeganeh et al. [13] carried out numerical analyses of a high-rise building behind an anchored wall, observing more damage in the model where the real building was simulated instead of being simplified as surcharge loading. Although the dynamic behaviour of retaining walls was investigated by many researchers previously [6–8], very little research exists in the literature that investigated the dynamic performance of the retaining walls in the presence of a structure on the backfill side, in close proximity to the retaining wall.

In many cases, the soil on the backfill side, as well as in front of the retaining wall, may be saturated, such as in the case of water-front structures. Earthquake loading can lead to full or partial liquefaction of such soils. In this paper, the focus is on dry sandy soils and the performance of the retaining wall is studied for this relatively simple soil case, so that the effect of the structure on the backfill can be easily identified. The presence of pore fluid in the saturated soil case would lead to additional loading due to hydrodynamic action on the retaining wall, as well as degradation in soil stiffness due to excess pore pressure generation.

Overall, this paper adopts a novel approach where an identical flexible retaining wall system will be analysed with and without a structure placed on the backfill side. This allows for a direct comparison between the retaining wall on its own and the changes induced by the presence of the structure and its foundations.

## 2. Numerical Modelling

In this paper, finite element modelling is carried out using the Swandyne code [9], which is a general-purpose FE code for problems in Geomechanics. As explained earlier, two models are analysed with and without a structure behind the retaining wall. The dimensions of the model with a structure are shown in Figure 2. The retaining wall had a retained soil height (*h*) of 3.6 m, and the wall itself penetrated to a depth (*d*) of 5.76 m, giving an *h/d* ratio of 0.625. The soil was modelled as a uniform dry sand bed on either side of the retaining wall. The structure was modelled as a sway frame with a high centre of gravity. In addition, loading on the roof was chosen to be asymmetric, with larger loading applied to the columns closer to the retaining wall. The columns of the structure were supported on individual strip footings. The right-hand side structural footing was placed about 2.1 m from the back edge of the retaining wall. In Swandyne FE code, two distinct steps are required for any dynamic analyses. Initially, a static run must be executed to establish the geostatic stresses in the soil and for the model to establish equilibrium of forces. In the second step, the final stress state from the static run is utilised as the initial stress state to start the dynamic run.

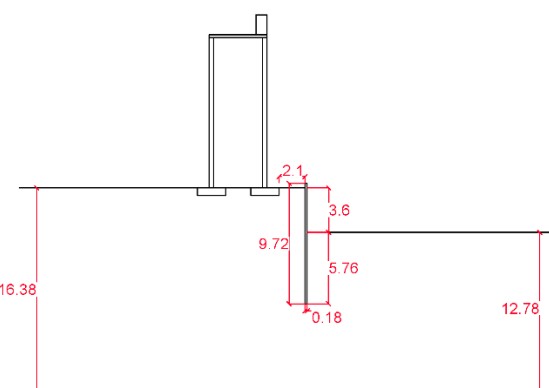

**Figure 2.** Dimensions of a retaining wall system (unit: m).

*2.1. Finite Element Discretisation*

The finite element discretisation used in the analyses described in this paper was carried out using eight noded isoparametric, quadrilateral elements with nine gauss points for integration. This enabled accurate modelling of the soil elements, as well as the structural elements forming the retaining wall and the

sway frame structure. Slip elements were utilised next to the retaining wall on both the active and passive sides and below the structure. The material properties used in these analyses are presented in Section 2.2.

An overview of the FE discretisation is shown in Figure 3a for the case of a retaining wall only and Figure 3b for the case of a retaining wall with a sway frame structure behind it. In each of these FE meshes, there were 1690 and 1716 solid nodes, respectively, with two degrees of freedom per node. The number of elements in each of these meshes were 781 and 789, respectively. The element sizes and time steps in the dynamic analyses were chosen to allow for the transmission of $S_h$ waves from the bedrock towards the ground surface following the recommendations of Semblat et al. [14] and Haigh et al. [15].

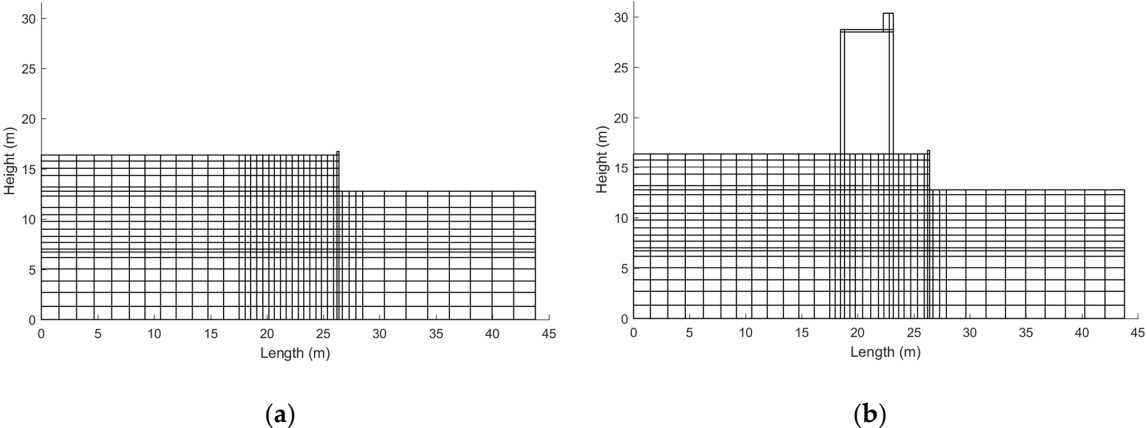

|   |   |
|:---:|:---:|
| (**a**) | (**b**) |

**Figure 3.** Finite element (FE) discretization: (**a**) a retaining wall only and (**b**) a retaining wall with an adjacent structure on the backfill.

In these FE analyses, the base nodes of the meshes shown in Figure 3a,b were fixed in both the $x$ and $y$ directions. This was necessary, since the earthquake loading was applied as nodal accelerations at all the fixed nodes in the meshes. In these analyses, a simple sinusoidal motions and sine sweep motions were used as the input accelerations to the base nodes. The reason for this choice of input motion is to determine the evolution of the peak bending moments at a time instant and the evolution of horizontal earth pressures next to the retaining wall, with and without the structure on the backfill. The edge boundaries were modelled as fixed in the $x$ direction and free in the $y$ direction. While these boundary conditions are satisfactory for the static analyses, it is recognised that unwarranted wave reflections may occur when earthquake loading is applied. In previous research, e.g., Madabhushi and Zeng [7,8] and Cilingir et al. [10], absorbing boundaries were used in the numerical analyses to simulate the boundaries in the dynamic centrifuge tests. In this paper, the lateral extent of the boundary value problem was sufficiently far from the active and passive zones of the boundary. The material and Rayleigh damping in the numerical analyses were considered to be large enough to absorb the wave reflections. Nonetheless, it must be pointed that some error will occur in the dynamic analyses due to the wave reflections from the vertical boundaries.

The flexural properties and densities of different components used in these analyses are presented in Table 1. The flexural stiffness is expressed for the retaining wall and the structural columns and is expressed as "per m", as the analyses were carried out with a plane strain formulation. The average density of the slab was calculated to allow for the additional mass added to the right-hand edge of the structure (Figure 3b) to provide additional structural loads for the right-hand side columns. This allows for the inclusion of a degree of asymmetry in the structure. The resulting bearing pressures on the left and right footings are also included in Table 1.

**Table 1.** Structural properties.

| Structural Element | Flexural Stiffness (EI) (MNm$^2$/m) | Density (kg/m$^3$) | Bearing Pressure (kPa) |
|---|---|---|---|
| structural columns | 66.68 | 2800 | - |
| slab | Rigid | 5906.7 | - |
| strip footing (left) | Rigid | 2800 | 83.0 |
| strip footing (right) | Rigid | 2800 | 117.0 |
| retaining wall | 34.02 | 2800 | - |

## 2.2. Constitutive Models

In the analyses described in this paper, three types of constitutive models were used. The simple "elastic" model was used for the sway frame structure including the roof slab and the footings. Similarly, the retaining wall was also modelled as an "elastic" material. The constitutive parameters ascribed to these materials are presented in Table 2. Interface elements were used between the retaining wall and the soil both on the active and passive sides. These were modelled as "slip elements", with the properties shown in Table 2.

**Table 2.** Details of constitutive parameters.

| Parameter | Value | | | Definition |
|---|---|---|---|---|
| | Dry Sand | Interface Elements | Structure/Retaining Wall | |
| Constitutive model | Mohr-Coulomb V | Slip | Elastic | Type of constitutive model used |
| Young's modulus | 50 MPa | 50 MPa | 70 GPa | Soil stiffness for static equilibrium |
| Young's modulus (dynamic) | 50 MPa | 50 MPa | 70 GPa | Soil stiffness for damping in dynamic analyses |
| Poisson's ratio | 0.3 | 0.3 | 0.15 | Links strains in horizontal and vertical directions |
| Uniaxial yield stress | 100 Pa | - | - | Cohesion |
| Friction angle (critical state) | 30° | 16.4° | - | To obtain critical state failure line |
| Dilatancy angle | 2° | - | - | To obtain the peak friction angle |
| Work-hardening modulus | 100 | - | - | The slope of the stress vs. yield strain |
| Void ratio | 0.8 | - | - | For the calculation of material density |

For modelling soil behaviour in FE analyses of geotechnical problems, it is important to account for the plastic behaviour of soil. Accordingly, many elasto-plastic models are used in the FE analysis. One of the simplest soil models is the Mohr-Coulomb type model. Such a model has been adapted to handle the elastic-perfectly plastic behaviour of soil, which has been implemented in the FE code SWANDYNE. Various versions of such models are available in this code, with increasing levels of complexity. The analyses presented in this paper were carried out using Mohr-Coulomb V model. The main features of the model are briefly outlined below, and the actual parameters that are used are presented in Table 2.

Since the primary concern in these analyses is with dry, sandy soil, the nonassociated flow rule must be used. Many researchers have established that sandy soils follow the nonassociative flow rule [16–18]. The Mohr-Coulomb V model in SWANDYNE captures the nonassociative Mohr-Coulomb elastic-perfectly plastic response. The Mohr-Coulomb V constitutive model was used for cyclic loading problems investigated by previous researchers [7,10,19]. The main advantages of this constitutive model are discussed below:

- The variation of bulk and/or shear modulus is considered with mean confining effective stress.

- Cohesion can be included. The stress state will be cut off if the mean effective confining stress is more negative than the allowable cohesion. In the analyses presented in this, only a nominal cohesion of 100 Pa was used.
- The plastic potential can have a different slope with the yield surface.
- A smooth fit to the triaxial compression and triaxial extension state, so there is no corner or singularity in the $\pi$-plane [20].
- Strain hardening of the soil is incorporated in this model, so that the hardening of soil that occurs cycle by cycle during an earthquake load can be captured.

Some of the limitations of the Mohr-Coulomb model are given below:

- As the Mohr-Coulomb model is a simple model, the shapes of the yield surface and plastic potential surface are similar.
- This model cannot be used for saturated soil conditions, as it does not capture the volumetric strains under cyclic loading, and hence, no excess pore pressure build-up occurs.

In the present analyses, a nominal cohesion of 100 Pa was used primarily to achieve numerical stability. Additionally, the interface friction between the retaining wall and the dry sand was chosen to be about 0.5 the peak friction angle of the sand.

In this paper, only uniform dry sand was considered. However, in case of waterfront structures, the shallow portions of the backfill above the water table can be under suction. Ng et al. [21] discussed the variation of soil stiffness and its influence on the retaining walls under such conditions. Although it is beyond the scope of this paper, it would be interesting to consider the influence of such suction dependent shear modulus and its variation under cyclic loading.

## 3. Static Equilibrium of the Retaining Wall

As explained earlier, before Swandyne code can be used to run any dynamic analyses, it requires a static run to establish the geostatic stresses. Such runs were carried out for the present analyses, and in this section, the static behaviour of the retaining wall on its own will be compared to the case where a structure is present on the backfill side. It must be pointed out that, in running the static analyses, the retaining wall was wished-in-place, i.e., the construction process of the retaining wall was not taken into account.

### 3.1. Wall Deflections and Bending Moments

The two important parameters in the retaining wall design are the static wall deflections on completion of the wall construction and the bending moments that are generated in the wall section. In Figure 4, the normalised static wall deflections are plotted against the normalised depth. Both these parameters are normalised with the total height of the wall $H_{wall}$ used in these analyses. In Figure 4, the normalised wall deflections obtained for the case of the retaining wall only (i.e., "no structure") are compared with the case when the sway frame structure was present on the backfill side (see Figure 3b). Clearly, the wall deflections increased substantially when the structure is in place behind the retaining wall. The wall tip deflection increased nearly by a factor of three.

In Figure 4, the increased curvature of the wall can also be seen. This implies the bending moments in the wall, which are proportional to the curvature, should also increase. In Figure 5, the normalised bending moments are plotted against the normalised depth. The bending moments $M$ are normalised by the unit weight of the soil and wall height ($H^3$) to obtain a nondimensioned variable. The same normalisation will be used for the results from dynamic analyses presented later. In Figure 5, it can be seen that the normalised bending moments increased substantially in the presence of the structure. Further, it can also be seen that the location of the peak bending moment was just below the excavation depth for the retaining wall-only case. This location of peak bending moment seems to shift downwards in the presence of the structure in the backfill. In general, the shapes of the bending moment curves are as expected in a static analysis.

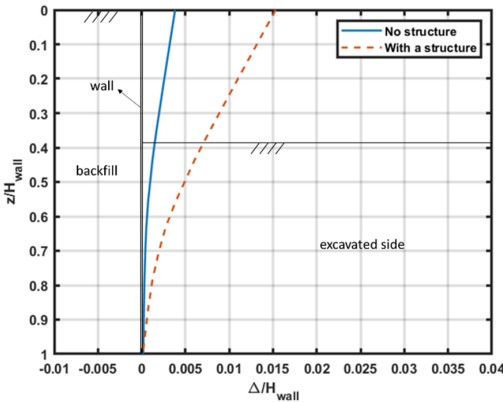

**Figure 4.** Static deflection of the retaining wall.

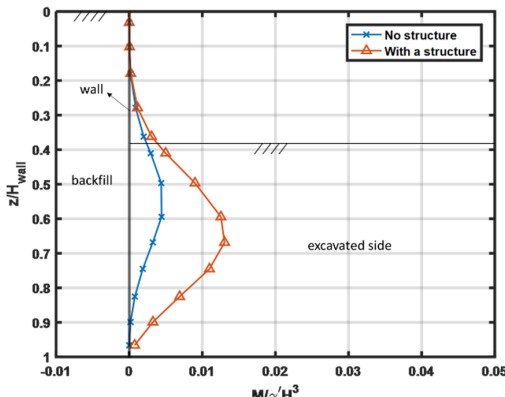

**Figure 5.** Normalised static bending moments in the retaining wall.

### 3.2. Horizontal Stresses and Earth Pressures

It would be interesting to compare the horizontal stresses mobilised in the soil for the two cases. In Figure 6a, the horizontal stress contours in the soil body are plotted for the "retaining wall-only" case in the unit of kPa. In this figure, it can be seen that, generally, the horizontal stresses increase with depth. However, there is a bulge of increased horizontal stress in front of the retaining wall on the passive side, which is required to establish the equilibrium of the retaining wall. There are also small drops in the horizontal stresses on the active side. In Figure 6b, an equivalent contour plot for the horizontal stresses in the case with the structure on the backfill side is presented. In this figure, the horizontal contours are not very uniform on the active side due to the presence of the two footings of the structure. The horizontal stresses below the footings are expected to increase due to the increased bearing pressure of the footings, causing horizontal stress bulbs to form in this region. It is also interesting to see that, on the passive side, the horizontal stresses increase just in front of the wall. A much larger pressure bulb is formed in this region than in the "retaining wall-only" case. Again, this larger pressure bulb is required to establish equilibrium in the "structure on backfill" case.

In addition to the general distribution of the horizontal stresses shown in Figure 6a,b, it is possible to obtain the horizontal earth pressures acting at each gauss point of the soil elements next to the retaining walls from the FE analyses. In addition, the Coulomb's earth pressure coefficients can be calculated using the following well-known Equations (1) and (2) for active and passive cases:

$$K_a = \left( \frac{\frac{\sin(\alpha - \varphi)}{\sin \alpha}}{\sqrt{\sin(\alpha + \delta)} + \sqrt{\frac{\sin(\varphi + \delta)\sin(\varphi - \beta)}{\sin(\alpha - \beta)}}} \right)^2, \tag{1}$$

$$K_p = \left( \frac{\frac{\sin(\alpha+\varphi)}{\sin\alpha}}{\sqrt{\sin(\alpha-\delta)} - \sqrt{\frac{\sin(\varphi+\delta)\sin(\varphi+\beta)}{\sin(\alpha-\beta)}}} \right)^2. \tag{2}$$

In Figure 7, the static distributions of the earth pressures acting on the retaining wall for the "wall-only" case and the "structure on the backfill" case are plotted, both on the active side and the passive side. The theoretical earth pressure distributions given by Equations (1) and (2) are also plotted in this figure. On the active side, the earth pressures for the "wall-only" case are pretty close to the Coulomb's active earth pressure line. However, for the "structure on the backfill" case, the active earth pressure shows a pressure bulge between 2–4-m depth. This is due to the increased horizontal pressures coming from the foundations of the structure. On the passive side, for the "wall-only case", it can be seen that the passive pressures do not fully mobilise compared to the Coulomb's distribution. This is well-known, and only partial passive pressures are mobilised to attain equilibrium. The FE analyses also show that the passive earth pressure mobilisation depends on the vertical stress and the lateral strain. Passive earth pressures are larger when larger strains are mobilised due to wall deflection and reach a peak at about a depth of 5.5 m. Below this depth, they start to decrease as the mobilised strains reduce.

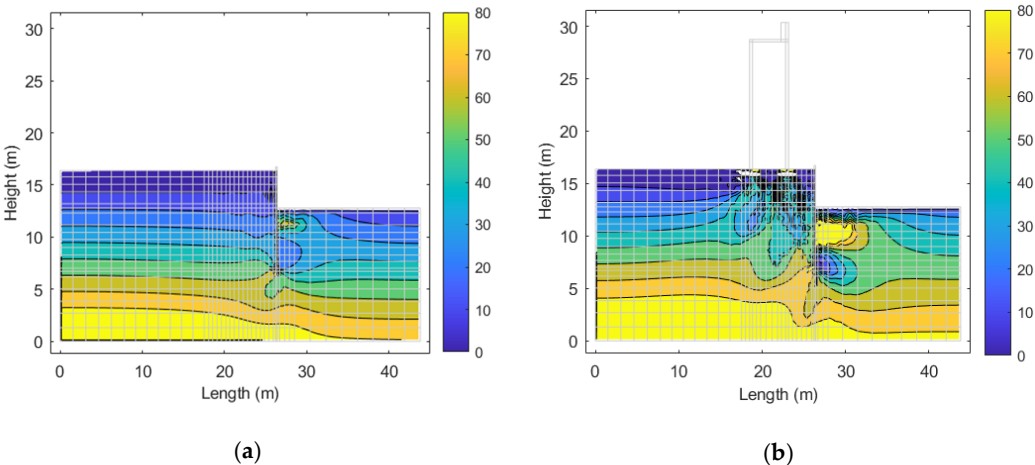

(**a**)　　　　　　　　　　　　　　　　　　　　　　　(**b**)

**Figure 6.** Horizontal stresses in the soil for: (**a**) the "'retaining wall-only" case and (**b**) the "structure on the backfill" case.

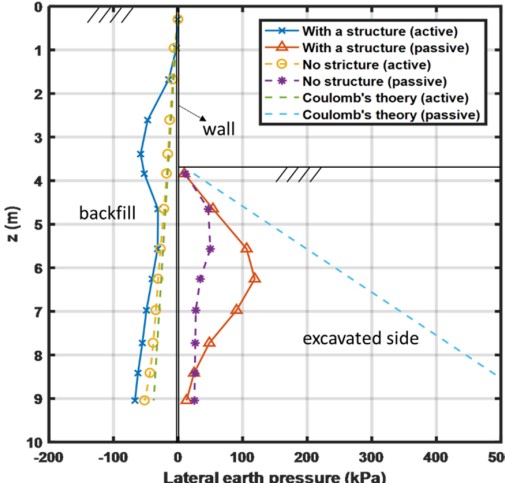

**Figure 7.** Horizontal earth pressures acting on the retaining wall.

In the case of "structure on the backfill", much larger passive pressures are mobilised, as required to establish the horizontal static equilibrium. This is consistent with the bending moments observed earlier in Figure 5. As seen in Figure 7, even in this case, the passive pressures are much smaller than the Coulomb's passive pressure line given by Equation (2).

## 4. Dynamic Response of the Retaining Wall

Following the static runs, dynamic analyses were carried out using the same constitutive models and parameters shown in Table 2. The final stress state from the static run was used as the initial stress state in the dynamic runs. Several different earthquake loadings were considered, such as a sine sweep motion and sinusoidal motions of different amplitudes, to replicate moderate and strong earthquakes.

### 4.1. Acceleration Time Histories

In Figure 8, the accelerations recorded at the top of the retaining wall are presented, along with the input motion applied at the base nodes. This input motion was a 1-Hz sinusoidal motion with 10 cycles of loading. It has a peak acceleration of about 0.23 g. The input motion was obtained from a dynamic centrifuge test conducted earlier at 60 g using the servo-hydraulic earthquake actuator [16]. In this figure, the wall top accelerations are presented for the cases of "wall-only" and "structure on the backfill". As expected, the peak accelerations are larger for the case with the structure on the backfill. Additionally, in both cases, the input accelerations are amplified by the time they reach the wall top. The amplification is larger when the structure is present on the backfill.

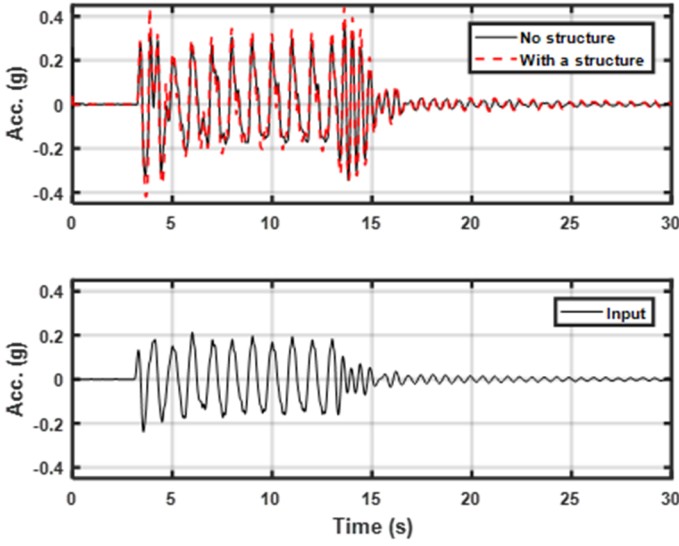

**Figure 8.** Wall top accelerations.

### 4.2. Dynamic Bending Moments

The additional loading applied by the earthquake causes dynamic bending moments in the retaining wall. In Figure 9, the envelopes of the maximum normalised bending moments are plotted against normalised depth for the cases of "wall-only" and "structure on the backfill" subjected to the same input motion shown in Figure 8. In Figure 9, it can be seen that the dynamic bending moments are much larger for the case of the retaining wall with a structure on the backfill. It is also possible to compare this figure with the static bending moments shown in Figure 5 and conclude that the dynamic bending moments are much larger. In fact, they increased by a factor of almost 2.5 during this 0.23-g earthquake for the case with a structure on the backfill. In addition, the location of the peak bending moment appears to have shifted deeper relative to the static case.

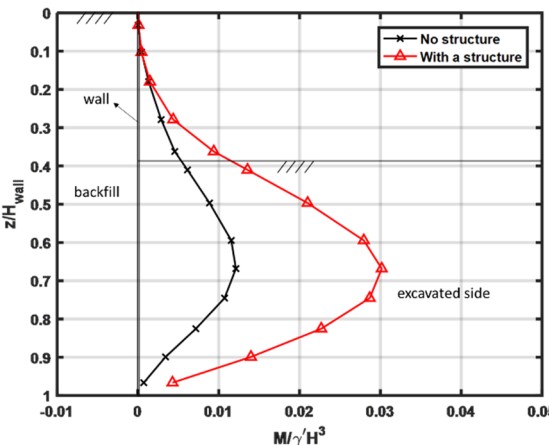

**Figure 9.** Envelopes of a maximum, normalised dynamic bending moment.

However, the peak bending moments may not occur at the same time at all elevations of the retaining wall. Therefore, in Figure 10, the normalised bending moment profiles are plotted at different time instants marked on the input motion plotted at the bottom of the figure. Time instants 1 and 3 were chosen at 0 acceleration (maximum +ve and −ve velocity), and time instants 2 and 4 were chosen at maximum +ve and −ve accelerations (zero velocity).

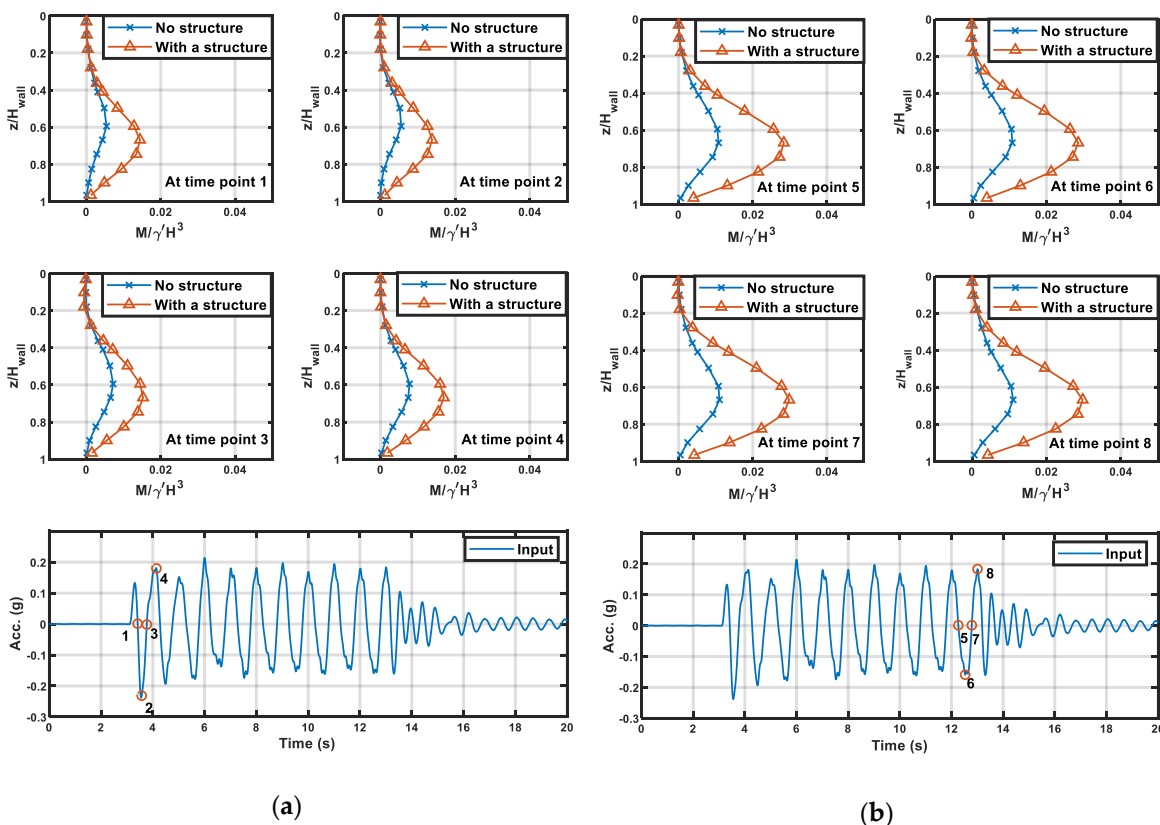

**Figure 10.** Dynamic bending moment profiles at different time instants: (**a**) during an earlier cycle and (**b**) during a later cycle.

In Figure 10a, the bending moment variations through time instants 1 to 4 during an earlier cycle for both cases of "wall-only" and "structure on the backfill" are presented. There is no great reversal of bending moments, as the retaining wall is locked into a particular curvature due to the soil. However, the peak bending moment does increase from time point 1 to 4, with small drops in value at time

points 2 and 3. This is due to the direction of soil acceleration and ensuing changes in dynamic earth pressure. The accumulation of bending moments can continue with the increasing number of loading cycles throughout the earthquake loading. This is illustrated in Figure 10b, in which the bending moment variations are plotted for a later loading cycle at time points 5–8. In this figure, it can be seen that the peak bending moments for both cases (i.e., with and without the structure on the backfill) increased relative to the earlier cycle presented in Figure 10a. The retaining wall can accrue more deflections away from the backfill during the dynamic loading but cannot move back into the soil. This is why the bending moments can either increase when deflection of the wall increases (and, hence, the curvature of the wall increases) or can remain the same but can never decrease by a significant amount.

## 5. Soil Response

Accelerations can be obtained at any of the solid nodes in the FE discretisation shown in Figure 3a,b. In Figure 11, the acceleration-time histories obtained in a soil column in the free field, far away from the retaining wall, are presented. The depth of the node below the ground surface is shown in each plot. It can be seen that the input accelerations amplify as they propagate upwards to the ground surface. Additionally, there is very little difference between the "wall-only" case and the "structure on the backfill" case, supporting that this column represents the far-field assumption reasonably well. Similar acceleration-time history plots can be obtained for the nodes on the active side and passive side. It may be more instructive to plot the amplification factor defined as the peak acceleration at a node normalised by the peak input acceleration. Such a plot is presented in Figure 12 for the free-field, active and passive sides.

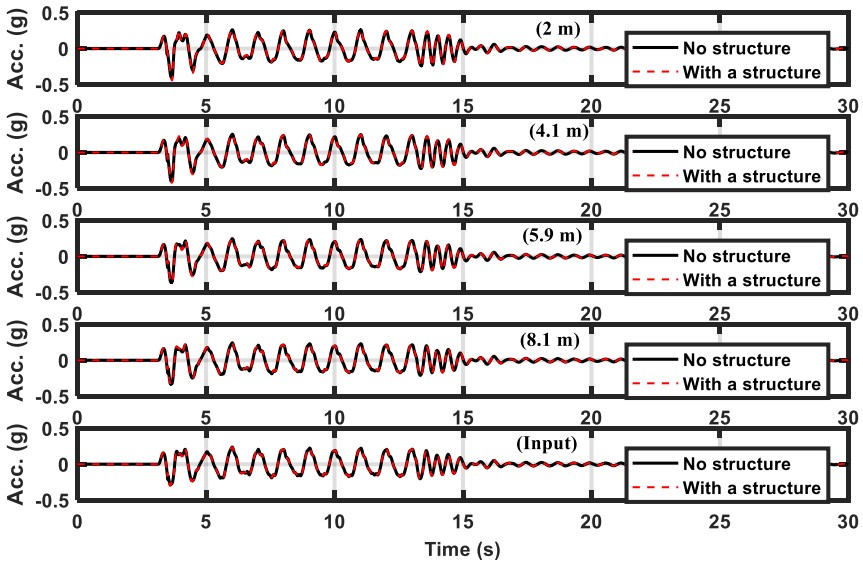

**Figure 11.** Acceleration time histories in the free-field soil.

Referring to Figure 11, the amplification factor increases from 1 at the bottom to about 1.5 at the ground surface in the free field. Such amplification in dry, relatively loose sand layers has been previously observed both in earthquake case histories, as well as other numerical analyses [7,8]. In the active zone behind the retaining wall, a difference in amplification factors is noticed for the case of "structure on the backfill". The amplification that occurs towards the ground surface is almost completely stopped by the presence of the structure. This may be due to the additional vertical stresses applied by the footings increasing the confining pressures in this zone, forcing the soil to act more like a rigid body. Similarly, on the passive side, a small attenuation of the amplification factors is noticed. It is not clear why the presence of structure should influence the soil response on the passive side, and this aspect may require further investigation.

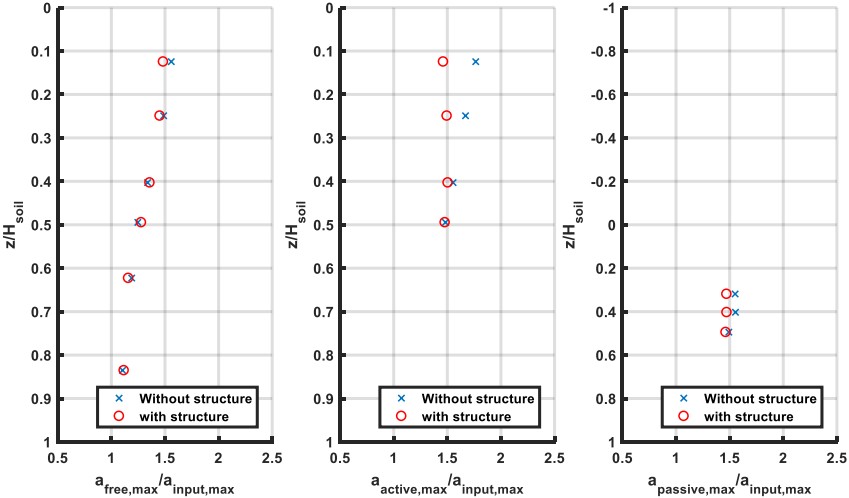

**Figure 12.** Amplification factor in the free-field, active and passive sides.

## 6. Dynamic Response of the Structure

One of the objectives of this paper is to investigate the response of the structure placed adjacent to a retaining wall on the backfill. The response of the structure was investigated first by applying a sine sweep input motion in the range of 0.1–2 Hz. The peak acceleration magnitude of this input motion is only 0.05 g. As in the case of the sinusoidal motion, this input motion was taken from a previous dynamic centrifuge test of Madabhushi et al. [22]. In Figure 13, the input motion and its Fast Fourier transform (FFT) are plotted at the bottom. The acceleration response of the roof of the sway frame and its FFT are plotted at the top in Figure 13. Due to the differential settlement of the footings as a consequence of the lateral displacement of the retaining wall described earlier, the structure also experiences vertical movements, which are also shown in Figure 13. It can be seen that the magnitude of these vertical accelerations is quite small and more towards the higher-frequency end. Moreover, it can be seen that the frequencies above 0.25 Hz are significantly amplified by the time they reach the roof slab. There is also a sharp spike in the amplification at 1 Hz, which is the natural frequency of the sway frame.

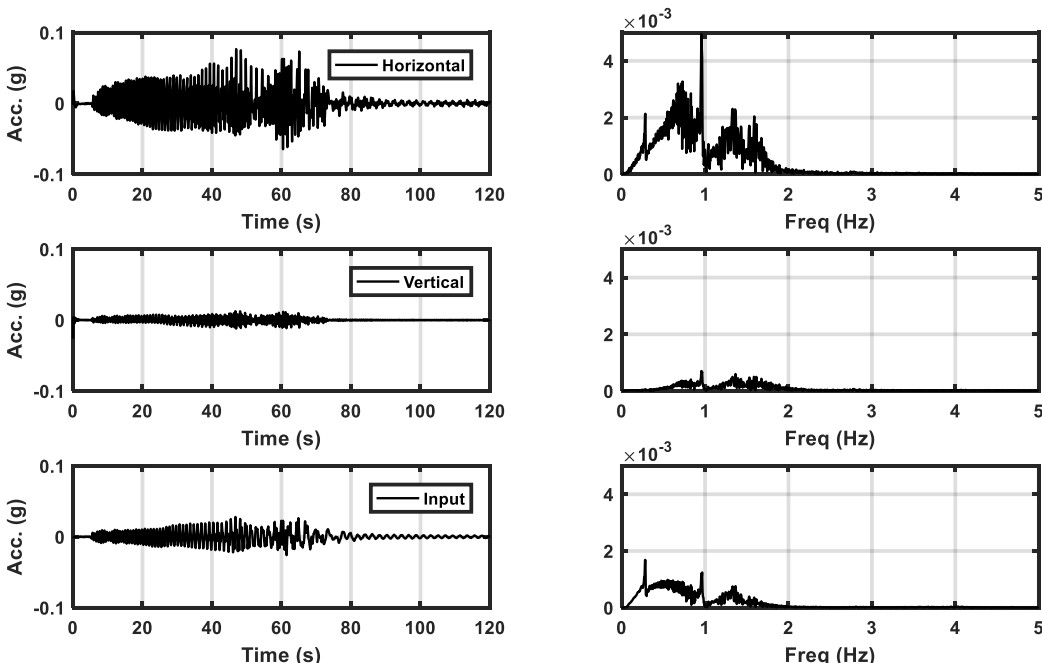

**Figure 13.** Structural response to a sine sweep motion.

The dynamic response of the structure is also investigated during a stronger 1-Hz sinusoidal earthquake motion introduced earlier. In Figure 14, the input acceleration and the horizontal and vertical accelerations recorded on the roof slab are presented. The input acceleration in this case has a maximum value of 0.23 g. However, as this is applied at 1 Hz, which is the natural frequency of the sway frame, large accelerations were registered by the roof slab. In the horizontal direction, these accelerations reach a magnitude of nearly 0.4 g, while, in the vertical direction, a peak acceleration of 0.1 g was registered. The FFT plots in Figure 14 also confirm the amplification of accelerations at 1 Hz.

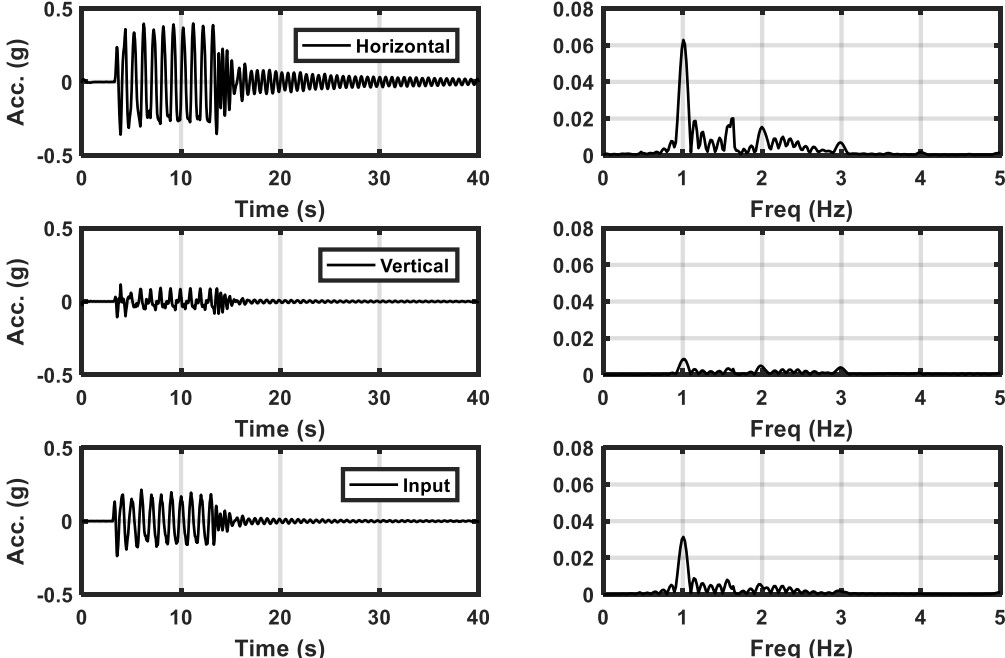

**Figure 14.** Structural response to a sinusoidal motion.

Another interesting observation that was obtained from these dynamic analyses was the rotation suffered by the structure. The structure suffered rotation due to the outward movement of the retaining wall creating a settlement trough in the backfill and a consequent differential settlement between the left and right footings. In Figure 15, the rotation suffered by the structure obtained from the differential settlement of the footings is plotted, along with the sinusoidal input motion. In this figure, it can be seen that the structure accumulates rotation in each cycle of the input motion and accrues a total, residual settlement of about 4°. It is also interesting to see that the rotation ceases to accrue after the end of the strong cycles in the input motion.

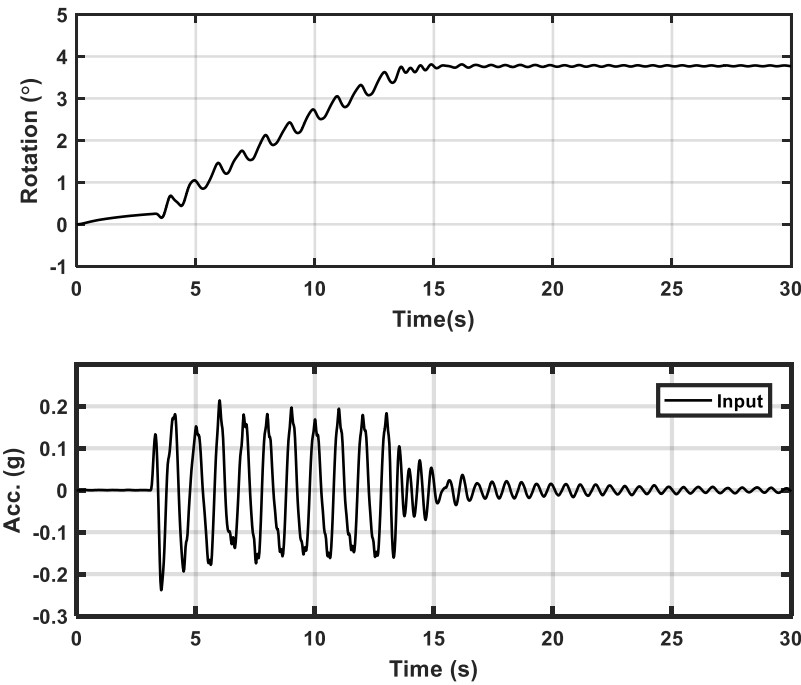

**Figure 15.** Rotation of the structure due to a sinusoidal input motion.

## 7. Dynamic Earth Pressures

The earthquake loading will induce additional dynamic earth pressures on the retaining wall. Traditionally, limit equilibrium methods such as the Mononobe-Okabe (M-O) method developed by Mononobe [23] and Okabe [24] are used to calculate the dynamic lateral earth pressures. In the M-O method, the dynamic inertial forces are considered as additional static forces applied to the soil wedge. The dynamic active and passive earth pressure coefficients can be calculated by modifying the Coulomb's equations given earlier in Equations (1) and (2), as follows:

$$K_{ae} = \frac{\cos^2(\varphi - \alpha - \theta)}{\cos(\theta)\cos^2(\alpha)\cos(\delta + \alpha + \theta)\left[1 + \sqrt{\frac{\sin(\varphi + \delta)\sin(\varphi - \beta - \theta)}{\cos(\delta + \alpha + \theta)\cos(\beta - \alpha)}}\right]^2}, \tag{3}$$

$$K_{pe} = \frac{\cos^2(\varphi + \alpha - \theta)}{\cos(\theta)\cos^2(\alpha)\cos(\delta - \alpha + \theta)\left[1 + \sqrt{\frac{\sin(\varphi + \delta)\sin(\varphi + \beta - \theta)}{\cos(\delta - \alpha + \theta)\cos(\beta - \alpha)}}\right]^2}, \tag{4}$$

and noting that

$$\tan\theta = \frac{k_h}{(1 - k_v)}. \tag{5}$$

Using the above equations, it is possible to have the limiting dynamic earth pressure lines as shown in Figure 16, along with the maximum dynamic earth pressures obtained from the dynamic analyses for the sinusoidal input motion with a peak acceleration of 0.23 g. It should be pointed out that the M-O method will result in an increased active pressure line and a reduced passive pressure line relative to the static earth pressures predicted by Coulomb's equations shown in Figure 7. It can be seen that the dynamic passive earth pressures are much larger in the case of the structure on the backfill compared to the "wall-only" case. These are also larger than the static earth pressures for both cases, with the peak horizontal earth pressure reaching nearly 200 kPa for the case with a structure on the backfill. This is an increase of nearly 50 kPa compared to the static earth pressure for this case. For both cases, the dynamic passive earth pressures are lower than those predicted by the M-O method, although this line on the passive side is closer to this limiting earth pressure line, compared to the

static earth pressures to the Coulomb passive pressure line shown in Figure 7. However, the dynamic active earth pressures on the backfill side obtained by the numerical analyses are larger than those predicted by the M-O line for both the "wall-only" and "structure on the backfill" cases. The bulge in active earth pressure at about 2.75-m depth attributed earlier to the presence of the structure still persists in the dynamic analyses.

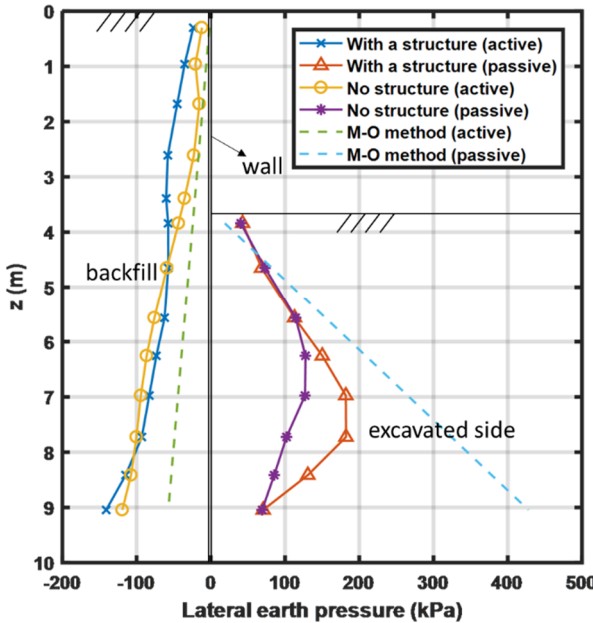

**Figure 16.** Distribution of maximum horizontal earth pressures acting on the retaining wall. M-O: Mononobe-Okabe.

In addition to the maximum horizontal earth pressures, it is possible to trace the evolution of these earth pressures through a cycle of earthquake loading. In Figure 17a, the horizontal earth pressures are plotted on the active and passive sides at four different time instants, 1–4, for an early cycle. These time instants are noted on the input acceleration at the bottom and were explained earlier with respect to Figure 10. Unlike the bending moments, the horizontal earth pressures can increase and decrease depending on the earthquake loading direction. For example, following the passive earth pressure for the "structure on the backfill" case, it can be seen that, between time instants 1 and 2, the peak horizontal earth pressure increased from 100 to 140 kPa. However, between time instants 3 and 4, this earth pressure decreases from 130 to 105 kPa. Similar observations can be made for the "wall-only" case. Additionally, similar increases in active earth pressure can be seen on the backfill side in Figure 17a. For comparison, the horizontal earth pressures are plotted for a later cycle at time instants 5–8, as shown in Figure 17b. It is interesting to note that the magnitude of earth pressures is much larger for this later cycle of earthquake loading, particularly on the passive side. For example, for the structure in the backfill case, the magnitude of peak horizontal earth pressure increased from 125 kPa to 180 kPa for time instants 2 and 6, respectively (i.e., peak negative acceleration points). This increase in horizontal earth pressure may be due to the accumulation of locked-in stresses cycle by cycle and partly due to the increase in soil density resulting from earthquake-induced settlements. The location of the peak horizontal earth pressure shifted to a deeper elevation in this later cycle. Similar observations can be made for other time instants as well. On the active side, the increases are much less pronounced. However, the active earth pressures for the "no structure" case seem to have increased more than the "with structure" case in this later cycle, especially at deeper elevations. This aspect requires further investigation. It should be noted that, in all cases, the M-O predictions underestimate the horizontal earth pressure on the active side.

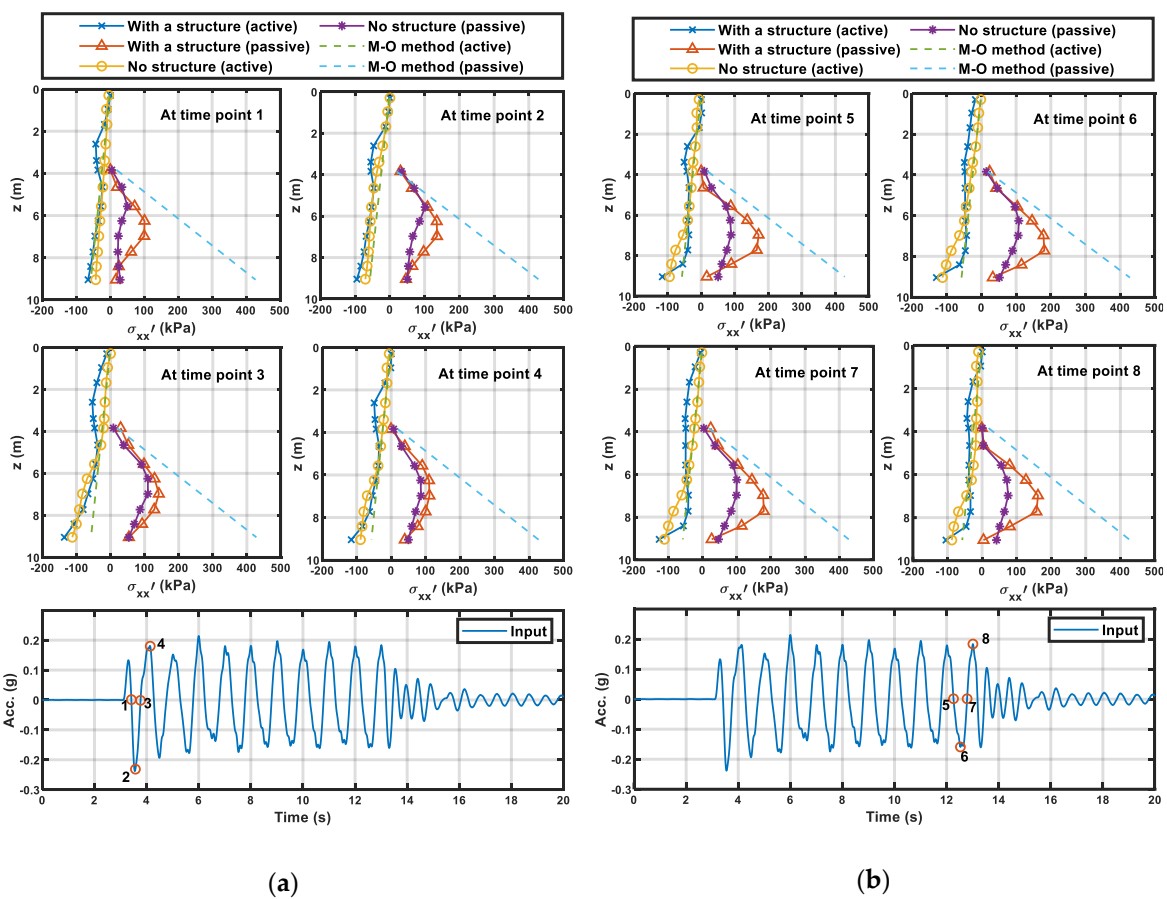

**Figure 17.** Evolution of the horizontal earth pressures through: (**a**) an earlier earthquake loading cycle and (**b**) a later earthquake loading cycle.

## 8. Conclusions

In this paper, static and dynamic analyses were carried out on a retaining wall in dry sand with an *h/d* ratio of 0.625. These were compared to a case where a sway frame structure was placed on the backfill adjacent to the retaining wall. The main objective of this research was to investigate the strong soil-structure interaction that may occur due to the presence of the structure. Using the static analyses, it was shown that the deflections and bending moments in the retaining wall increased due to the presence of the structure on the backfill. In addition, the horizontal pressure bulb that developed in front of the retaining wall also increased in size when the structure was present on the backfill. Further, the horizontal earth pressures obtained from these analyses were compared to Coulomb's predictions.

Using the dynamic analyses, the response of the top of the retaining wall was found to increase in the presence of the structure on the backfill. The maximum bending moments due to the earthquake loading increased in magnitude. By considering the bending moments at different time instants, it was shown that these bending moments locked in and did not reduce in value. The dynamic response of the soil was considered next, and it was shown that the amplification ratio below the structure on the backfill was smaller than for the "wall-only" case, suggesting more "rigid body"-like accelerations below the footings owing to the increased confining stress. The structural response was considered next, and the large amplification of accelerations of the roof slab was explained. Finally, the dynamic earth pressures acting on the retaining wall were considered both in terms of their maxima, as well as their evolution with time. Unlike the bending moments, it was shown that the dynamic earth pressures can both increase and decrease depending on the direction of the input accelerations. The maximum dynamic earth pressures were compared to the traditional M-O method. It was seen

that the M-O method overpredicts the dynamic passive earth pressures, while it underpredicts the active earth pressures.

**Author Contributions:** X.G. carried out the numerical analyses reported in this paper, with help from the second author. G.S.P.M. helped prepare the first draft of this manuscript, while the first author created all the figures shown in this paper. All authors have read and agreed to the published version of the manuscript.

**Funding:** This research received no external funding.

**Acknowledgments:** The first author would like to thank the financial support from China Scholarship Council and Cambridge Trust for her PhD studies.

**Conflicts of Interest:** The authors declare no conflict of interest.

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
