# Peer review of "Numerical Modelling of Structures Adjacent to Retaining Walls Subjected to Earthquake Loading"

_geosciences, doi:10.3390/geosciences10120486_

Round 1

Reviewer 1 Report

The article describes a set of numerical analyses of the static and dynamic response of a structure adjacent to a retaining wall. There are at least four major issues with this paper as listed below:

1- The manuscript is suffering from a major flaw which is the lack of calibration and verification of the numerical model. Any numerical model (e.g. finite element, finite difference, discrete element, etc.) should be calibrated and verified prior to use for a research study without an exception.
So the question to the authors is that how did they verified their model and can prove that their results are reliable?

2- It appears that this work is a presentation of a case history of a certain building constructed adjacent to a retaining wall (Figure 1) during a hypothetical earthquake scenario. But no information is presented on the details of such case history. So what are the bases for this unique retaining wall system plus the structure? How did you decide on the soil properties? How did you decide on structural properties? How did you choose the dimensions of the structure and the retaining wall? and so on and so forth.

3- It is an accepted method to subject numerical simulation to an earthquake histogram that is recorded in the vicinity of the subjected area and published by the governments and universities. But it is not acceptable to use a harmonic excitation for earthquake geotechnical analyses. This imposes a considerable amount of limitations to the results of the simulation, rendering the outcome useless for real-life projects and models.

4- Another major limitation is the use of the sole constitutive model of dry sand in this simulation. Since this is a purely numerical investigation, a variety of soils should be tested under drained and undrained conditions to provide some useful content to research work.

Reviewer 2 Report

This study carries out a numerical study on the behaviour of a retaining wall system with and without a structure placed on the backfill side. The results are useful for other researchers and engineers working in the similar field and have great implications to the practical engineering problems. The paper is well written. This reviewer has some comments that may help the authors to further improve the manuscript. The paper can be accepted for publication when all the raised comments can be addressed satisfactorily. The general and detailed comments are listed as follows.

  1. Please highlight the scientific gap or novelty of this work. What do you mean by “a novel approach” in L63?
  2. The adopted soil model is Mohr-Coulomb model. This model is actually over-simplified. Please discuss its advantages and shortcomings when modelling such soil-structure interaction problems by using Mohr-Coulomb model.
  3. If unsaturated soil mechanics is considered in this study, e.g., suction-dependent soil shear modulus, what will be the changes in the predicted results? Please comment on this by referring to the paper “Use of unsaturated small-strain soil stiffness to the design of wall deflection and ground movement adjacent to deep excavation” . Will the results be more conservative from the design perspectives?
  4. The current work is purely based on numerical studies. Is it possible to add a few field data to some figures to have the comparison between measured and computed results, so that the readers are more convinced?

Reviewer 3 Report

A well-written paper discussing results from simulations of a cantilever retaining structure subjected to static and dynamic earthquake loading with and without a structure present on the retained side. Please address the following comments.

  1. In a dynamic loading situation like an earthquake, the behaviour of soil is often dictated by the generation of pore water pressure (positive or negative). Please include in the discussion why no pore water pressure or fluid mechanical interaction was not considered in the analysis?
  2. Please comment on capability of Mohr-Coulomb V model to capture the behaviour of granular soils subjected to cyclic loading.
  3. The results presented are for the first loading cycle. It will be interesting to compare the observation at a later loading cycle - Fig 10.
  4. In the literature review section (line 52-62), please briefly outline some of the observations made by the research works cited
  5. Line 130 to 140 – please paraphrase. Reads like directly copied from Madabhushi and Zeng (2007)
  6. Please cite the source for figure 1. Also, mention the location of this construction.
  7. Why the density of slab material was taken as much larger than other parts of the structure (Table 1)?

Madabhushi, S.P.G., and Zeng, X. 2007. Simulating Seismic Response of Cantilever Retaining Walls. Journal of Geotechnical and Geoenvironmental Engineering 133(5): 539-549. doi: doi:10.1061/(ASCE)1090-0241(2007)133:5(539).

Reviewer 4 Report

In this paper, numerical results are presented related to a case study of a flexible cantilever sheet pile wall with and without a structure on the backfill side. The aim of the numerical investigations is of studying the effects of the interaction on the retaining wall by considering also the influence due to the structure on the backfill side. The paper is well organized and the Reviewer believes that it may be published after a revision, following the comments below reported. 1. In the introduction please highlight what’s the novelty of this study. In addition, it should be considered that the interaction problems among transit loads – backfill - retaining wall is well known also in the case of spandrel walls of existing masonry bridges, acting as retaining walls for the incoherent backfill. This consideration would make the work presented much more fascinating in the published literature. Among the others, about the interaction transit loads-backfill-spandrel walls, the following studies may be considered: D’Amato, M., Laterza, M., Casamassima, V. M.,. 2017. “Seismic Performance Evaluation of Multi-Span Existing Masonry Arch Bridge”. The Open Civil Engineering Journal Suppl-5 (M11), 1191–1207. https://doi.org/10.1063/1.4992619. Pelà, L., Aprile, A., and Benedetti, A., 2009. "Seismic Assessment of masonry arch bridges". Eng. Struct., 31 (8), 1777-1788. Marcheggiani, L., Clementi, F., Formisano, A. 2020. “Static and dynamic testing of highway bridges: a best practice example”. Journal of Civil Structural Health Monitoring, 10 (1), 43-56. https://doi.org/10.1007/s13349-019-00368-1. 2. About the numerical models. It is not clear how the case study was chosen. Moreover, how were defined the dimensions of the mesh adopted. Please, clarify these aspects. 3. A better description of the Figure 4 and Figure 5 should be done. 4. It is not clear how were selected the time histories for the dynamic analyses. In addition, please specify where these inputs were applied in the numerical models. This is important for understanding how the spatial motion of the signal could have influenced the response. 5. In the conclusion the most important remarks of what may be generalized from the case study analyzed should be added.

Round 2

Reviewer 1 Report

The revised manuscript is slightly clearer than the previous one. However, after reading the revised paper and going over the concerns raised by the other 3 reviewers and the authors’ responses to all comments (from all 4 reviewers), the following issues still remain and are not addressed:

  • It is very good that the software (in this case SWANDYNE) is used before, and previous researchers have shown its usefulness. However, the model used by the authors is still not calibrated. The Cilinger model is not the same as the model that the authors used. In other words, a researcher(s) can only use a model for numerical analysis study of affecting parameters if they calibrate their model (in this case the FE model of the retaining wall with a structure) using actual data from either lab or field, and only then to use it to vary parameters and investigate the change in behaviors.
  • I still did not see any justification whatsoever for any of the dimensions and soil parameters in this work. It is very good that authors are conducting more wholesome research on the topic, but even for the larger study, they need to present justifications for everything. The audience cannot be referred to future publications in order to understand this work.
  • use of sinusoidal wave, single soil (dry sand ), and a single constitutive model makes this work extremely limited. Most of the conclusions that are made in this work based on these parameters are known matters and are expected. For instance “Unlike the bending moments, it was shown that the dynamic earth pressures can both increase and decrease depending on the direction of input accelerations.”, and “The maximum dynamic earth pressures were compared to the traditional M-O method. It was seen that the M-O method over-predicts the dynamic passive earth pressures, while it under-predicts the active earth pressures” are not new discoveries. Simply put, as long as authors are using single parameters in their analysis, the conclusions are only limited to those conditions that are used in the calculations and do not have any physical meaning in real-life engineering.

Reviewer 2 Report

The authors have addressed my concerns satisfactorily. The paper can be accepted now.

Author Response

No further comments were made and the authors would like to express thanks to the reviewer.

Reviewer 4 Report

The paper was improved and it can be accepted for publication 

Author Response

(The authors gave the same response as above.)
